# Polymorphisms in the Drug Transporter Gene *ABCB1* Are Associated with Drug Response in Saudi Epileptic Pediatric Patients

**DOI:** 10.3390/biomedicines11092505

**Published:** 2023-09-11

**Authors:** Rania Magadmi, Reem Alyoubi, Tahani Moshrif, Duaa Bakhshwin, Bandar A. Suliman, Fatemah Kamel, Maha Jamal, Abdulhadi S. Burzangi, Sulman Basit

**Affiliations:** 1Department of Clinical Pharmacology, Faculty of Medicine, King Abdulaziz University, Jeddah 21589, Saudi Arabia; thani_moshrif@hotmail.com (T.M.); dbakhshwin@kau.edu.sa (D.B.); foakamel@kau.edu.sa (F.K.); mjamal@kau.edu.sa (M.J.); burzangi@kau.edu.sa (A.S.B.); 2Pediatric Department, Faculty of Medicine, King Abdulaziz University, Jeddah 21589, Saudi Arabia; raalyoubi@kau.edu.sa; 3Clinical Pharmacy Department, King Abdullah Medical Complex, Jeddah 23816 , Saudi Arabia; 4College of Applied Medical Sciences, Taibah University, Madinah 42353, Saudi Arabia; bsuliman@taibahu.edu.sa; 5Biochemistry and Molecular Medicine Department, College of Medicine, Taibah University, Madinah 42353, Saudi Arabia; sbasit.phd@gmail.com; 6Centre for Genetics and Inherited Diseases, College of Medicine, Taibah University, Madinah 42353, Saudi Arabia

**Keywords:** anti-seizure medications, drug response, pediatric neurology, pharmacogenetic, polymorphism

## Abstract

Epilepsy is one of the most common chronic neurodisorders in the pediatric age group. Despite the availability of over 20 anti-seizure medications (ASMs) on the market, drug-resistant epilepsy still affects one-third of individuals. Consequently, this research aimed to investigate the association between single-nucleotide polymorphisms (SNPs) of the ATP-binding cassette subfamily B member 1 (*ABCB1*) gene in epileptic pediatric patients and their response to ASMs. This multicentric, cross-sectional study was conducted among Saudi children with epilepsy in Jeddah, Saudi Arabia. The polymorphism variants of *ABCB1* rs1128503 at exon 12, rs2032582 at exon 21, and rs1045642 at exon 26 were genotyped using the Sanger sequencing technique. The study included 85 children with epilepsy: 43 patients demonstrated a good response to ASMs, while 42 patients exhibited a poor response. The results revealed that good responders were significantly more likely to have the TT genotypes at rs1045642 and rs2032582 SNPs compared to poor responders. Additionally, haplotype analysis showed that the T-G-C haplotype at rs1128503, rs2032582, and rs1045642 was only present in poor responders. In conclusion, this study represents the first pharmacogenetic investigation of the *ABCB1* gene in Saudi epileptic pediatric patients and demonstrates a significant association between rs1045642 and rs2032582 variants and patient responsiveness. Despite the small sample size, the results underscore the importance of personalized treatment for epileptic patients.

## 1. Introduction

Epilepsy is one of the most prevalent serious chronic neurological disorders. The worldwide prevalence of active epilepsy is approximately 6.38 per 1000 people, and the global incidence of epilepsy is more than 67 per 100,000 persons per year [1]. Despite the availability of twenty anti-seizure medications (ASMs), about one-third of epileptic patients have refractory epilepsy (i.e., seizures not controlled by two or more appropriately selected drugs) [2]. Therefore, drug resistance is a common clinical problem in the management of epilepsy. The development of drug resistance, inadequate seizure control, and intolerable side effects are common issues encountered during the course of epilepsy treatment. These limitations necessitate the importance of treatment change in the management of epilepsy. Treatment change, including switching to alternative ASMs or adjusting the dosage regimen, is a critical strategy employed to improve seizure control and optimize patient outcomes. Moreover, early identification of the need for treatment change and timely intervention can prevent disease progression and minimize long-term complications associated with uncontrolled seizures [2]. Therefore, understanding the factors influencing treatment change in epilepsy is of paramount importance in providing optimal care for epileptic patients.

According to the pharmacokinetic hypothesis, ASMs do not reach the epileptic target site in the concentrations necessary for therapeutic efficacy, partly as a result of active efflux, which is mediated by locally overexpressed drug transporter proteins in brain tissue. The drug efflux transporter P-glycoprotein (Pgp) is encoded by the multidrug-resistant transporter gene adenosine triphosphate (ATP)-binding cassette subfamily B member 1 (*ABCB1*), which was first discovered in cancer cells that had developed resistance to anti-cancer drugs and is known to be a multispecific drug transporter [3].

Many factors related to P-glycoprotein are affected by various single-nucleotide polymorphisms (SNPs) present in the *ABCB1* gene, for example, its expression, efflux, substrate specificity, and mRNA stability.

In order to test the concept of *ABCB1* SNPs affecting response to ASMs, numerous studies in Caucasian and Asian populations (Japanese, Korean, and Han Chinese) have been conducted [4,5,6]; nevertheless, the findings have been conflicting and inconclusive. The number of studies supporting a relationship with drug resistance phenotype is comparable to the number of studies that have failed to support this relationship. Moreover, because there may be significant differences in gene frequencies and their effects in different ethnic populations, it is necessary and important to explore this relationship (positive or negative) among other ethnic groups. Based on the hypothesis that *ABCB1* genetic variants can modulate drug response phenotype in epilepsy, this study was carried out to assess the association of three SNPs of the *ABCB1* gene, rs1128503 at exon 12, rs2032582 at exon 21, and rs1045642 at exon 26, and the drug response to ASMs in Saudi epileptic children. Relevant previous studies reported the associations between these polymorphisms and anti-seizure response in different populations [4,5,6]. However, this study was conducted specifically on the Saudi pediatric population due to several reasons. Firstly, Saudi Arabia has a relatively high prevalence of epilepsy, with a reported incidence of 5.9 per 1000 person-years [7]. Secondly, ethnic differences in drug response have been reported previously, and it is important to understand the genetic factors that may influence drug response in this population. Thirdly, there is a lack of data on the association between *ABCB1* genetic variants and drug response in Saudi pediatric patients, and our study aims to fill this gap in knowledge.

## 2. Materials and Methods

### 2.1. Ethical Approval and Ethical Consideration

This study’s protocol was approved by the Unit of Biomedical Research Ethics Committees in the Faculty of Medicine, at King Abdulaziz University with approval number (Reference No 44–22). Written informed consent was required prior to starting to collect the participants’ data or samples. For patient privacy and security, each participant was given a special code rather than a patient name. Participants were free to quit at any time.

### 2.2. Participants and Study Design

The study was a multicenter, hospital-based, cross-sectional study of epileptic pediatric patients visiting pediatric neurology clinics during the period from September 2020 to December 2020. Demographic and clinical data were collected from patient medical records.

During the patients’ clinic visits, a pediatric neurology consultant assessed the medical history of all patients including their: gender, etiology of epilepsy, medication history, past medical history, and history of development of liver function test (LFT) disturbance. Then, the pediatric neurology consultant evaluated those who met the inclusion criteria. Inclusion criteria for this study involved: patients aged from 2 to 17 years old, diagnosed with epilepsy, and receiving ASM treatment for at least one year. Additionally, patients were required to have normal psychometric functions and normal neurologic examination. Finally, the legal caregiver should agree to participation. In contrast, exclusion criteria for this study were patients lacking records, patients who did not have reliable seizure frequency, patients who were not compliant with ASM treatment, patients with liver disorders, and patients whose caregiver did not provide written consent or did not visit the clinic regularly.

Then, the participants’ caregivers were asked to read and sign the informed consent form after the primary investigator explained the research’s purpose and process to them. Another co-investigator was a witness as well. If the participants’ caregivers approved participation in the study, patients were classified as good responders or poor responders according to the definition set by the International League Against Epilepsy (ILAE) [8]. Patients were classified as poor responders if the treatment with different ASMs as a monotherapy or in combination, correctly prescribed for at least 12 months, at maximally tolerated doses, failed and epileptic seizures persisted. Meanwhile, good responders were classified as those who were totally free from seizures for at least 1 year during treatment with ASMs as a monotherapy or in combination at optimal tolerated therapeutic doses.

### 2.3. Genotyping

#### 2.3.1. Deoxyribonucleic Acid (DNA) Extraction and Quantification

At the end of the visit, five milliliters of blood were collected from the patients into sterile EDTA tubes for DNA analysis. The blood samples were stored at 4 °C until the genomic DNA extraction procedure was performed. A total of 200 μL of peripheral blood samples was used. Genomic DNA was purified with a QIAamp DNA Mini Kit (Catalogue # 51306, Qiagen, Alameda, CA, USA). In brief, 20 μL of Qiagen Proteinase K was pipetted into the bottom of a 1.5 mL microcentrifuge tube. Then, 200 μL of blood was added to the Proteinase K. Finally, 200 μL of buffer AL (for lysis) was added to the tube and vortexed for 15 s. Then, the tube was incubated in a 56 °C water bath for 10 min. Next, the tubes were briefly centrifuged to remove any droplets that may have formed at the top. Then, 200 μL of 100% ethanol was added and mixed by vortexing for 15 s. Then, the mixture was carefully transferred to a QIAamp spin column. Then, the tube was centrifuged at 8000 rpm for a minute. This step allows the binding of genetic materials to silica gel inside the spin column. After that, the collection tube was discarded, and then QIAamp spin column was placed into a new collection tube. Then, 500 μL of buffer AW1 (washing) was added. After that, the tubes were centrifuged at 8000 rpm for a minute. Then, 500 μL of buffer AW2 (washing) was added and centrifuged again at 14,000 rpm for 3 min. After that, the QIAamp spin column was transferred to a sterile 1.5 mL microcentrifuge tube. Then, 50–200 μL of AE buffer was added to the tubes and then centrifuged for 2 min at 8000 rpm. The final step was repeated to have an extra DNA tube. DNA was then kept at 4 °C overnight. The DNA concentration (ng/mL) and the quality and quantity of DNA were measured using a NanoDrop™ 2000c spectrophotometer (Thermo Scientific, USA). DNA samples with an optical density at an A260/A280 ratio of 1.8–2.0 were processed for further analysis.

#### 2.3.2. Polymerase Chain Reaction (PCR)

Prior to sequencing, PCR amplification for all DNA samples was carried out to confirm the presence and size of PCR products. The PCR reaction was prepared by adding 1 μL of forward and reverse primers (100 pM, pH 8), 1 µL of template DNA, 8 µL of dH_2_O, and 10 µL of Hot Began HotStart Green Taq Master mix (Canvax™). Thermal cycling was performed on Veriti 96-Well Thermal Cycler (Applied Biosystems, Waltham, MA, USA) with an initial denaturation at 94 °C for 1 min, followed by 40 cycles of denaturation at 94 °C for 35 s, annealing at 58 °C for 35 s, and elongation at 72 °C for 1 min. A final elongation step was utilized at 72 °C for 7 min. The selected forward and reverse primer sequences used in this study are listed in Table 1.

Gel electrophoresis was performed on all PCR amplification reactions for all samples to assess the integrity of PCR amplification before progressing to DNA sequencing. Five μL of each PCR reaction was loaded onto 1.5% agarose gel using the M12 Complete Electrophoresis Package (Edvotek Inc., Washington, DC, USA) for 40 min at 90 V. The bands were visualized under UV light using the ChemiDoc-It Imaging System (Analytik Jena, Jena, Germany). A purification step was performed to remove excess enzymes, nucleotides, and primers from the PCR products.

#### 2.3.3. Sanger Sequencing

The PCR products that were purified as described above were used as a DNA template for Sanger sequencing. Optimal annealing temperatures, cycling conditions, and reagent concentrations for the PCR were determined using the Veriti 96-Well Fast Thermal Cycler (Applied Biosystems, San Francisco, CA, USA), which is equipped with a temperature gradient block. The denaturation temperature and duration were 94 °C and 30 s, respectively. Annealing was set for 30 s at temperatures ranging from 59 °C to 55 °C, varying with the primer pair and template. The extension was carried out at 72 °C for between 60 and 90 s, depending on fragment length.

The BigDye Terminator v3.1 Cycle Sequencing Kit (Thermo Fisher Scientific, Waltham, MA, USA) was used for DNA sequencing. Two master mixes were prepared separately for the sequencing reactions in centrifuge tubes (Eppendorf, Hamburg, Germany). The contents were appropriately mixed by pipetting into separate sterile reagent reservoirs. Aliquots of 8.5 μL each were made, using a multichannel pipette (Eppendorf, Germany), of the forward master mix into the desired number of wells on sterile 96-well PCR plates, and the same was carried out for the reverse master mix on additional PCR plates. The plates were then sealed with foil covers and placed on crushed ice while being protected from direct light. Before beginning the sequencing reaction, 3.5 μL of the PCR product was added to each well in the forward and reverse plates containing the master mix using a multichannel pipette (Eppendorf, Germany). The plates were centrifuged in the VARISPIN 15 bucket centrifuge (Cryste, Gyeonggi-do, Republic of Korea) at room temperature for 1 min at 3600 rpm. The plates were loaded into the 3500 Genetic Analyzer (Thermo Fisher Scientific, USA) using the manufacturer-recommended settings. Geneious Prime software was used for alignment and identifying the sequence variants (Geneious Prime v.2013.0.1).

### 2.4. Statistical Analysis

Categorical variables of the patients are represented as percentages and frequencies. Pearson’s chi-square test was performed for categorical variables to compare the factor differences between the “good responders” and “poor responders” groups. Allele and genotype frequencies, Hardy–Weinberg equilibrium (HWE), analysis of association with a response variable based on logistic regression, multiple inheritance models (dominant and recessive), and analysis of association of haplotypes with the response were carried out using SNPStats [9]. The dominant model assumes that individuals with one or two copies of the minor allele have the same phenotype, while the recessive model assumes that individuals with two copies of the minor allele have a different phenotype than those with one or no copies of the minor allele. A *p*-value of <0.05 was considered significant, and 95% confidence interval and odds ratio (OR) were also calculated.

## 3. Results

### 3.1. Demographics and Classification of the Patients

Initial screening involved 200 patients. However, only 85 participants met the inclusion and exclusion criteria, accepted to participate in this study, and provided complete data and blood samples.

Out of the 85 epileptic patients, 61.2% were male. Around 93% of the cases did not declare a family history of epilepsy, although parental consanguinity was reported among a third of the study sample. Based on ASM classification, half of the patients showed a good response, as shown in Table 2.

### 3.2. Genotype Distributions and Allele Frequencies of ABCB1 Polymorphisms

Table 3 summarizes the genotype distributions and allele frequencies of the three selected *ABCB1* SNPs, namely rs1128503 on exon 12, rs2032582 on exon 21, and rs1045642 on exon 26. The TT genotype was the least frequent for all tested SNPs in this cohort.

Notably, the distributions of genotypes of all three tested SNPs were consistent with Hardy–Weinberg equilibrium proportions (*p* = 0.51, 1.0, 0.35 for *ABCB1* polymorphisms rs1128503, rs2032582, and rs1045642, respectively). Moreover, there were no significant deviations in any subgroups (all *p* > 0.05).

Next, the genotype distributions of this project were compared with those of other populations, as shown at https://www.ncbi.nlm.nih.gov/ (accessed on 28 June 2023).

Compared to other populations, the results show that there is ethnic variation between different populations. The genotype distribution of the rs1128503 SNP in this cohort is comparable to that of American and European populations, as shown in Figure 1A. Meanwhile, East and South Asian populations show more TT genotypes, contrary to the African population, which showed no TT genotypes. The same trends were noted regarding the rs2032582 SNP, as shown in Figure 1B.

However, the genotype distribution of the rs1045642 SNP in this cohort is comparable to that of American and East Asian populations, while European and South Asian populations show more TT genotypes, as shown in Figure 1C.

### 3.3. Associations between ABCB1 Gene Polymorphisms and Patient Response to ASMs

To evaluate the effect of SNPs on the drug response, comparisons among the three variants of *ABCB1* were conducted. As shown in Table 4, two out of three SNPs (rs2032582 and rs1045642) showed significant associations with drug responsiveness.

The genotype distributions and allele frequencies of rs1128503 SNP at exon 12 did not differ significantly between the good responders and poor responders.

Regarding the rs2032582 SNP at exon 21, genotype comparisons showed that the good responders were significantly more likely to have the TT genotype than the poor responders (OR = 7.07, 95% CI = 1.33–37.65, *p* = 0.035). The variant TT genotype increases the possibility of good response to ASMs by seven times. However, there were no significant differences in the patients’ response to ASMs among other genotypes related to this SNP.

Regarding the rs1045642 SNP at exon 26, the good responders were significantly more likely to have the TT genotype when compared with poor responders. Moreover, the frequencies of the T allele in good responders were significantly higher than in poor responders (*p* = 0.036 and *p* = 0.01, respectively). The variant TT genotype increased the possibility of good response to ASMs by almost five times, while the T allele increased the responsiveness to ASM by more than two-fold.

Because genetic variations that affect drug pharmacokinetics and response can be inherited in various ways, genotype distributions were analyzed based on the models of inheritance, as summarized in Table 5. The results showed that there were no significant differences in dominant and recessive models of the rs1128503 SNP. However, the rs1045642 and rs2032582 SNPs’ recessive models showed significant results. The variant allele homozygous TT genotype was significantly more common among good responders when compared to the other combinations.

### 3.4. Association between the Haplotypes of ABCB1 Polymorphisms and the Response to ASMs

Haplotype analysis was performed for *ABCB1* rs1128503, rs2032582, and rs1045642 SNPs. The D_0_ values for rs1128503, rs2032582, and rs1045642 were 0.81, 0.8, and 0.71, respectively. *p*-values were <0.001, which suggests that these polymorphisms were in strong linkage disequilibrium. Eight haplotypes were identified among the study population, as shown in Table 6. The major haplotype of *ABCB1* in this cohort was CGC (rs1128503, rs2032582, rs1045642).

Table 6 shows the haplotype frequencies of *ABCB1* SNPs among good and poor responders to ASMs. The results show that the TGC haplotype was only present in poor responders. Therefore, this haplotype was a risk factor for ASM resistance. Meanwhile, the CTT haplotype was only present in good responders; however, its frequency was less than 5%. The global haplotype association *p*-value was 0.0052.

### 3.5. Associations between ABCB1 Gene Polymorphisms and the Risk of Liver Function Disturbance Caused by ASMs among Epileptic Patients

Among all epileptic patients included in the genotyping, 40% of the patients suffered from liver toxicities and 60% reported no liver toxicity. As shown in Table 7, participants with the heterozygous CT variant of rs1128503 SNP were more likely to not develop LFT disturbance compared to other genotypes. However, it did not reach statistical significance.

Likewise, participants with the heterozygous CT variant of the rs1045642 SNP were more likely to not develop LFT disturbance compared to other genotypes. However, it did not reach the statistical significance.

## 4. Discussion

Epilepsy, a highly prevalent disease, affects nearly 50 million individuals world-wide. In a community-based study conducted by Al Rajeh et al. in 2001, it was found that in Saudi Arabia alone, there are 6.54 epileptic patients for every 1000 individuals [7]. These figures emphasize the significant health concern that demands proper attention and treatment. Disturbingly, individuals with epilepsy face a heightened risk of premature death compared to the general population. In a study conducted by Milroy in 2011, it was revealed that this risk can be up to three times higher [10]. It is, therefore, evident that epilepsy is not only a medical condition that leads to seizures but also profoundly impacts the lives and overall well-being of those affected.

Several factors may impact patients’ response to ASMs. Therefore, this study aimed to determine which patients would likely exhibit responses to ASMs based on their genetic characteristics. Early identification of patients’ response to ASMs has the potential to enhance cost-efficiency, save time, and improve patient well-being.

Many studies explore the role of genetic variation on the outcome of ASMs in terms of efficacy and safety. More recent studies have focused on genetic factors that may impact drug resistance, specifically exploring the role of genetic SNPs in modifying the pharmacokinetic and pharmacodynamic mechanisms of ASMs. Drug transporters, drug-metabolizing enzymes (such as cytochrome P450), and drug targets, including sodium channels, calcium channels, potassium channels, and GABA receptors, are among the multiple genes responsible for drug responsiveness [11]. Acknowledging the involvement of pharmacogenetics in ASM response is crucial for the future of personalized treatment for epileptic patients.

Patients with drug-resistant epilepsy (DRE) are often unresponsive to a variety of ASMs with differing mechanisms of action. One hypothesis proposes that changes in the expression of efflux transporters at the blood–brain barrier (BBB) can hinder ASMs from reaching a sufficient concentration in the brain, despite appropriate levels in the bloodstream [6]. Our findings support this hypothesis and suggest that certain variations in the *ABCB1* gene may influence medication response in epileptic patients.

The *ABCB1* gene encodes the Pgp efflux transporter protein, which plays a crucial role in drug transport across the BBB. Variations in this gene could affect drug bioavailability, potentially contributing to the interindividual differences observed in drug response [12]. Studying genetic variation in the *ABCB1* gene could offer valuable insights into the pathophysiology of DRE and help personalize treatment strategies.

In this study, the three most commonly reported SNPs of *ABCB1*, namely rs1128503 at exon 12, rs2032582 at exon 21, and rs1045642 at exon 26, have been studied for possible association with ASM responsiveness and safety.

The genotype and allele frequencies of the *ABCB1* rs1128503 SNP at exon 12 did not differ significantly between the good responders and poor responders in this cohort. Consistent with the present study, several previous studies failed to find an association between this SNP and ASM response among Jordanian [13] and American [14] epileptic patients. Remarkably, the genotype distribution of the rs1128503 SNP in this cohort is comparable to that of American populations, which could explain the similar results. However, other populations with different genotype distributions also showed no association between this SNP and ASM response, including Taiwanese [15], Korean [5], Japanese [6], and Indian [11] epileptic patients. This confirms the previous hypothesis based on meta-analysis results that the rs1128503 SNP is a silent SNP that does not result in any amino acid alteration in Pgp and consequently no alternation in the phenotype [16]. The extensive meta-analysis conducted by [16], which included 57 studies involving 12,407 patients, showed no association between rs1128503 SNP and ASM response, irrespective of study populations.

It is noteworthy to mention a study that was conducted on Saudi children with acute lymphoblastic leukemia, which revealed that there is no correlation between the effectiveness and safety of glucocorticoids and the rs1128503 SNP [17].

On the other hand, results of the rs2032582 SNP located at exon 21 demonstrated that patients who responded well to ASMs had a higher likelihood of having the TT genotype than other genotypes. This conclusion is consistent with previous research conducted on epileptic patients from Tunisia [18,19] and Iranian [20] epileptic patients.

On the contrary, the findings of this study contradict those of Seo et al. (2006) who observed an increased frequency of the TT genotype at the rs2032582 SNP in DRE Japanese patients [6]. Similarly, investigations conducted on Korean [5] and Indian [11] epileptic patients from varying ethnic groups provided no evidence of an association between the rs2032582 SNP and ASM response. The observed inconsistency in results could be attributed to a difference in rs2032582 SNP genotype distribution between this study and the Asian population, where the TT genotype is more common. A recent study has proposed that ethnicity is a determinant in the link between the *ABCB1* rs2032582 polymorphism and DRE. According to the study, the *ABCB1* rs2032582 polymorphism may ultimately heighten the risk of DRE in Asians but not in Caucasians [21].

The findings of this study on the rs1045642 SNP at exon 26 indicate a significant association between the TT genotype and T allele of the SNP and ASM responsiveness. These results remain significant when considering the recessive model of this SNP. These results are in line with previous studies [12,22,23,24,25]. Stasiołek et al. (2016) found that the C allele of the SNP may increase the incidence of DRE, while the T allele has a protective effect and is associated with ASM response in pediatric Polish epileptic patients, including 173 epileptic and 98 non-epileptic children [23]. On the same line, Zhao et al. (2020) recently reported that pediatric epileptic patients with the *ABCB1* rs1045642 TT genotype showed a significantly higher response to ASMs than those with CC and CT genotypes [25]. Siddiqui et al. showed the same results but in the other direction. They showed that the CC genotype of the rs1045642 SNP was more frequent among DRE patients than the TT genotype [12]. Furthermore, a meta-analysis by Li et al. (2015) revealed a significantly decreased risk of ASM resistance in Caucasian patients with the T allele of the rs1045642 variant [16].

The present study also yielded results that contradict previous studies [19,20,26]. Those studies showed that TT genotypes and the T allele of the rs1045642 SNP were significantly more frequent in poor response patients than in good response patients. Moreover, some other studies failed to find any association between the rs1045642 SNP and ASM response among different ethnic populations, such as Chinese [27], German [28], Turkish [29], and Indian [11] epileptic patients.

Taken all together, the association between the studied SNPs and response to ASMs remains uncertain, as revealed by the comparison of recent studies and previous findings. One plausible explanation for this discrepancy is the genetic diversity across different ethnic groups. Indeed, SNPs may have distinct frequencies and effects across populations, affecting drug metabolism, transport, or target binding. Nonetheless, other factors also contribute to the variations in the results, such as sample size, age group, the different criteria to classify ASM responsiveness, and the ASM treatment guidelines. It is worth highlighting that the three SNPs examined in this study, namely rs1045642, rs1128503, and rs2032582, may have a more meaningful impact as a haplotype, according to a recent report [19]. Haplotypes represent combinations of multiple SNPs that tend to be inherited together, reflecting the genomic diversity of ancestral populations. Therefore, investigating haplotypes rather than individual SNPs may provide more accurate and comprehensive information on the genetic determinants of ASM response.

In the field of pharmacogenetics, haplotype analysis of specific genetic variants is a powerful tool to identify potential associations with drug response. In this regard, investigating the haplotype frequencies of *ABCB1* SNPs among good and poor responders to ASMs can shed light on the genetic mechanisms involved in treatment resistance. Therefore, haplotype analysis of the selected *ABCB1* SNPs in this study indicated a high linkage disequilibrium between the three SNPs, suggesting that these loci functioned as a complex haplotype with a potential synergistic effect. Regarding the haplotype frequencies of *ABCB1* SNPs among good and poor responders to ASMs, the results showed that the TGC (rs1128503, rs2032582, rs1045642) haplotype was only present in poor responders, indicating a potential risk factor for ASM resistance. In contrast, the CTT (rs1128503, rs2032582, rs1045642) haplotype was only present in good responders.

Interestingly, a similar haplotype distribution was observed in a previous study conducted on an Asian population, where the TGC haplotype was more likely to be present in poor responders [15]. In contrast, a recent study that evaluated drug-resistant epilepsy in a Tunisian population showed that the haplotype combination CTT was significantly associated with poor response [19]. These conflicting results emphasize the need for further research to fully elucidate the complexities of ASM response in different populations.

Although rs1128503 did not exhibit a significant association with drug response, it appears to play a role in haplotype analysis. Siddiqui and colleagues (2003) posited that the rs1045642 variant is situated within the *ABCB1* gene, in a large block of LD in close proximity to other variants (such as the synonymous rs1128503 in exon 12 and/or the non-synonymous rs2032582 in exon 21), which could potentially alter Pgp transport activity [12]. Subsequently, the rs1045642 SNP is considered a silent variant that could influence the response to ASMs only if inherited in a haplotype block in LD with rs1128503 and/or rs2032582 SNPs. Again, this highlights the importance of considering haplotypes rather than individual variants in identifying genetic factors that contribute to the variability in response to ASMs.

Despite these discrepancies, the haplotype analysis conducted in this study provides valuable information on the pharmacogenetics of ASMs in the population of this study, which has been insufficiently studied in this context. Further studies are needed to confirm the findings and identify potential therapeutic targets for personalized medicine.

A large and growing body of literature has consistently reported the association between *ABCB1* SNPs and Pgp expression and function [24,30,31]. It is well established that the efflux of Pgp protein at the BBB plays a crucial role in regulating the concentration of ASMs in the brain and, consequently, the effectiveness of the therapy. Although our study did not directly measure Pgp expression, it suggests that patients with the TT genotype of rs2032582 and rs1045642 SNPs might experience a decrease in Pgp expression at the BBB. This hypothetical decrease could potentially result in a higher concentration of ASMs in the brain, potentially leading to an improved response to treatment. However, further investigation is required to directly measure Pgp expression and validate this hypothesis. Future studies should aim to elucidate the precise relationship between these SNPs, Pgp expression, and the therapeutic outcomes of ASMs.

P-glycoprotein is a drug efflux pump that is expressed in the liver and plays a key role in the elimination of many drugs, including ASMs [32]. Therefore, liver dysfunction may lead to altered drug metabolism and elimination, which may affect drug response in epilepsy patients. Another significant finding of this study is that individuals with the heterozygous CT variant at rs1128503 and rs1045642 are protected from liver toxicity associated with ASMs. This could be attributed to the variation in *ABCB1* SNPs leading to the modification of Pgp in hepatocytes, resulting in increased efflux of ASMs and safeguarding the liver [33]. It is worth mentioning that there is little information in the literature regarding studying the association of *ABCB1* SNPs and the development of ASM-induced liver toxicity among epileptic patients.

### Limitations and Strengths

Overall, these findings highlight the significance of *ABCB1* SNPs and ASM therapy outcomes and emphasize the need for individualized treatment plans tailored to a patient’s specific genotype. However, the sample size was relatively small in this study, especially for studying the haplotypes. Another limitation is that the serum levels of ASMs were difficult to measure due to the nature of the cross-sectional study. However, the study is crucial to support pharmacogenomic studies to identify the role of the human genome in the response to ASMs and provide significant clinical improvement in the management of drug-resistant epilepsy.

## 5. Conclusions

This study provides valuable insights into the use of pharmacogenetics to improve the therapeutic response in patients with epilepsy. Statistical analysis revealed a significant difference in *ABCB1* variants rs2032582 and rs1045642 between good responders and poor responders, with the T allele and TT genotype being more common in good responders.

These findings suggest that screening for these polymorphisms and haplotypes may be an effective method for personalized therapeutic management of epilepsy. By identifying the best treatment option for each patient and predicting the treatment outcome of newly diagnosed pediatric patients before the administration of medication, we can improve the effectiveness and safety of epilepsy treatment.

Overall, this study contributes to the growing body of research on personalized medicine, highlighting the potential of pharmacogenetics to improve therapeutic outcomes and healthcare delivery.

## Figures and Tables

**Figure 1 biomedicines-11-02505-f001:**
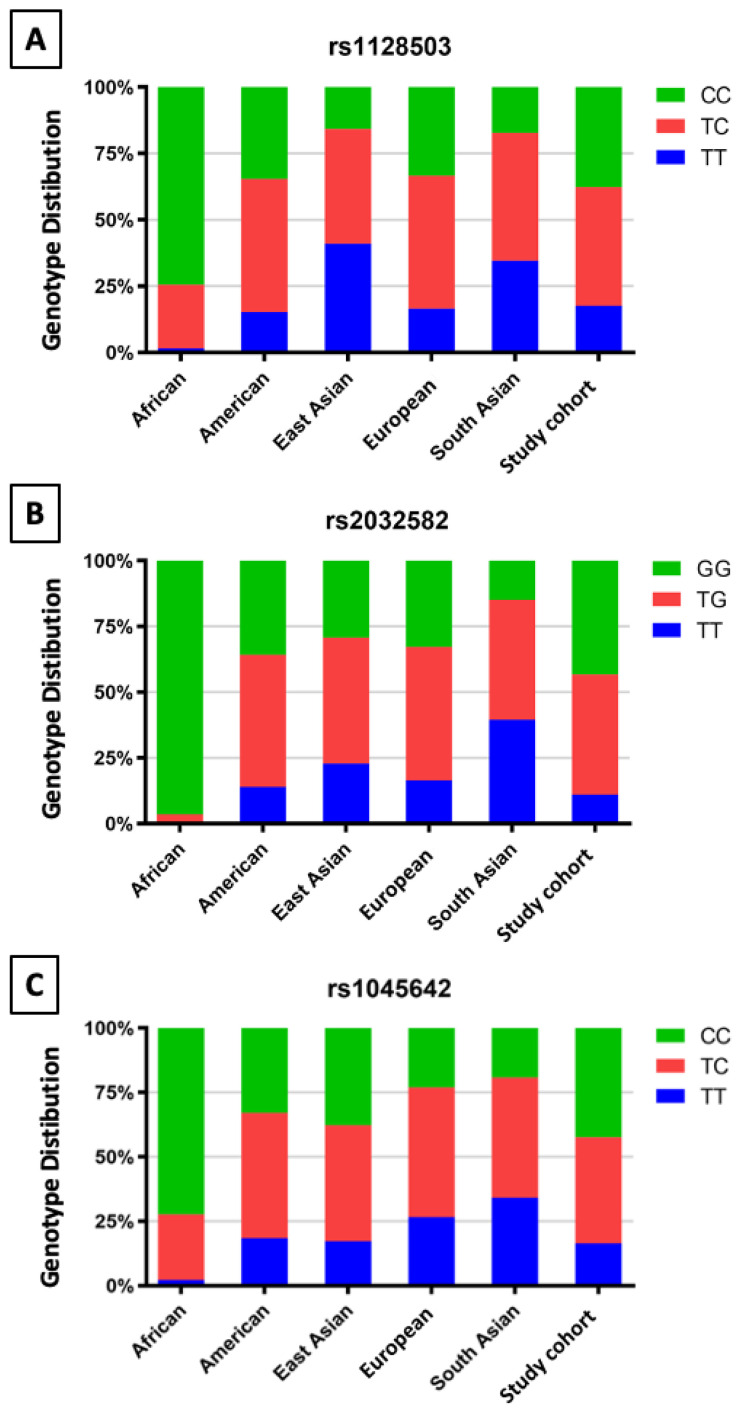
Genotype distributions and allele frequencies of *ABCB1* polymorphisms among different populations.

**Table 1 biomedicines-11-02505-t001:** The forward and reverse primer sequences of selected SNPs.

SNP ID	Exon Location	Nucleotide Change Location	Protein Change Location	Forward Primer	Reverse Primer
rs1128503	Exon12	C1235T	p. Gly412Gly	CCTGTGTCTGTGAATTGCCT	GTCATCTCACCATCCCCTCT
rs2032582	Exon21	G2677T	p. Ala893Ser	TCCTTCATCTATGGTTGGCAAC	AGCATAGTAAGCAGTAGGGAGT
rs1045642	Exon26	C3434T	p. lle1145lle	GAATGTTCAGTGGCTCCGAG	TGCTCCCAGGCTGTTTATTT

**Table 2 biomedicines-11-02505-t002:** Demographic and classification of the patients.

Characteristic		N (%)
Gender	Male	52 (61.2%)
	Female	33 (38.8%)
Parental consanguinity	Yes, 1st degree	28 (32.9%)
	No	57 (67.1%)
Family history of epilepsy	Yes	6 (7.1%)
	No	79 (92.9%)
Patient classification based on AED drug response	Good responder	43 (50.6)
	Poor responder	42 (49.4)

Data are presented as patient number (N) and percentage (%).

**Table 3 biomedicines-11-02505-t003:** Genotype distributions and allele frequencies of *ABCB1* polymorphisms among all patients (N = 85).

Genotype/Allele	N (%)
**rs1128503 at exon 12**	
CC	32 (37.6)
CT	38 (44.7)
TT	15 (17.7)
T	68 (40)
C	102 (60)
**rs2032582 at exon 21**	
GG	36 (42.4)
GT	38 (44.7)
TT	11 (12.9)
G	110 (64.7)
T	60 (35.3)
**rs1045642 at exon 26**	
CC	36 (42.3)
CT	35 (41.2)
TT	14 (16.5)
C	107 (63)
T	63 (37)

Data are presented as patient number (N) and percentage (%).

**Table 4 biomedicines-11-02505-t004:** Genotype and allele frequencies of *ABCB1* gene polymorphisms among good responders and poor responders to ASM (N = 85).

Genotype/Allele	Good Responders N (%)	Poor Responders N (%)	Adjusted OR (95% CI)	*p*-Value
	N = 43 (50.6)	N = 42 (49.4)		
**rs1128503 at exon 12**				
CC	15 (34.9)	17 (40.5)	1.00	0.38
CT	18 (41.9)	20 (47.6)	1.02 (0.40–2.62)	
TT	10 (23.3)	5 (11.9)	2.27 (0.63–8.14)	
T	38 (44)	30 (64)	1.00	0.26
C	48 (54)	54 (36)	0.702 (0.38–1.3)	
**rs2032582 at exon 21**				
GG	14 (32.6)	22 (52.4)	1.00	0.035 *
GT	20 (46.5)	18 (42.8)	1.75 (0.69–4.40)	
TT	9 (20.9)	2 (4.8)	7.07 (1.33–37.65)	
G	48 (55.8)	62 (73.8)	1.00	0.25
T	38 (44.2)	22 (26.2)	1.46 (0.76–2.81)	
**rs1045642 at exon 26**				
CC	14 (32.6)	22 (52.4)	1.00	0.036 *
CT	18 (41.9)	17 (40.5)	1.66 (0.65–4.27)	
TT	11 (25.6)	3 (7.1)	5.76 (1.36–24.36)	
C	46 (53)	61 (73)	1.00	0.01 *
T	40 (47)	23 (27)	2.31 (1.22–4.37)	

Data are presented as number of patients (N) and percentage (%). Data were analyzed using chi-square test. OR, odds ratio; CI, confidence interval. OR estimated by logistic regression analysis after adjustment for regimen. * *p* value < 0.05 considered significant.

**Table 5 biomedicines-11-02505-t005:** Comparison of different *ABCB1* genotype models of inheritance in poor ASM responders and good responders.

Model	Genotype	Good Responders N (%)	Poor Responders N (%)	Adjusted OR (95% CI)	*p*-Value
		N = 43 (50.6)	N = 42 (49.4)		
**rs1128503 at exon 12**					
Dominant	CC	15 (34.9)	17 (40.5)	1.00	0.59
	CT–TT	28 (65.1)	25 (59.5)	1.27 (0.53–3.06)	
Recessive	CC–CT	33 (76.7)	37 (88.1)	1.00	0.17
	TT	10 (23.3)	5 (11.9)	2.24 (0.69–7.24)	
**rs2032582 at exon 21**					
Dominant	GG	14 (32.6)	22 (52.4)	1.00	0.064
	TG–TT	29 (67.4)	20 (47.6)	2.28 (0.91–5.49)	
Recessive	GG–TG	34 (79.1)	40 (95.2)	1.00	0.021 *
	TT	9 (20.9)	2 (4.8)	5.29 (1.07–26.19)	
**rs1045642 at exon 26**					
Dominant	CC	14 (32.6)	22 (52.4)	1.00	0.064
	CT–TT	29 (67.4)	20 (47.6)	2.28 (0.95–5.49)	
Recessive	CC–CT	34 (74.4)	39 (92.9)	1.00	0.019 *
	TT	11 (25.6)	3 (7.1)	4.47 (1.15–17.40)	

Data are presented as number of patients (N) and percentage (%). Data were analyzed using chi-square test. OR, odds ratio; CI, confidence interval. OR estimated by logistic regression analysis after adjustment for regimen. * *p* value < 0.05 considered significant.

**Table 6 biomedicines-11-02505-t006:** Haplotype frequencies of ABCB1 polymorphisms among good and poor responders to ASMs (N = 85).

Haplotype			Haplotype Frequency		*p*-Value
rs1128503	rs2032582	rs1045642	Good Responder (N = 43)	Poor Responder (N = 42)	
C	G	C	41 (47.6)	49 (58.3)	0.26
T	T	T	32 (37.2)	17 (20.2)	0.19
C	G	T	2 (2.3)	4 (4.8)	0.57
T	G	C	0	7 (8.3)	<0.0001 *
T	G	T	5 (5.8)	2 (2.4)	0.24
T	T	C	1 (1.2)	4 (4.8)	0.35
C	T	C	4 (4.7)	1 (1.2)	0.31
C	T	T	1 (1.2)	0	<0.0001 *

Data are presented as number of patients (N) and percentage (%). Data were analyzed using two-tailed, paired *t-*test. * *p* value < 0.05 considered significant.

**Table 7 biomedicines-11-02505-t007:** Association between *ABCB1* gene polymorphisms and the risk of liver function test disturbance caused by ASMs among epileptic patients (N = 85).

Genotype	LFT Disturbance N (%)	No LFT Disturbance N (%)	Adjusted OR (95% CI)	*p*-Value
	N = 34 (40)	N = 51 (60)		
**rs1128503 at exon 12**				
CC	16 (47.1)	16 (31.4)	1.00	0.06
CT	10 (29.4)	28 (54.9)	2.80 (1.03–7.62)	
TT	8 (23.5)	7 (13.7)	0.87 (0.26–2.99)	
**rs2032582 at exon 21**				
GG	17 (50)	19 (37.3)	1.00	0.38
GT	12 (35.3)	26 (51)	1.97 (0.76–5.12)	
TT	5 (14.7)	6 (11.7)	1.42 (0.34–5.91)	
**rs1045642 at exon 26**				
CC	17 (50)	19 (37.2)	1.00	0.061
CT	9 (26.5)	26 (51)	2.58 (0.95–7.04)	
TT	8 (23.5)	6 (11.8)	0.67 (0.19–2.33)	

Data are presented as number of patients (N) and percentage (%). Data were analyzed using chi-square test. OR, odds ratio; CI, confidence interval. OR estimated by logistic regression analysis after adjustment for regimen. LFT, liver function test; ASMs, anti-seizure medications.

## Data Availability

The raw data that support the findings of this study are available from the corresponding author upon reasonable request.

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
