# Peer review of "Polymorphisms in the Drug Transporter Gene ABCB1 Are Associated with Drug Response in Saudi Epileptic Pediatric Patients"

_biomedicines, 2023, doi:10.3390/biomedicines11092505_

Round 1

Reviewer 1 Report

The manuscript was prepared very well. The introduction section justifies the purpose of the study. I congratulate the authors for the preparation of the manuscript

I would like to congratulate the authors for the structure of the manuscript and all the research carried out. It is highly publishable. However, there are some concerns, in part important, so the review articles need revision, see below.

Introduction

-        Why is this study considered relevant?

-        Why is this study necessary?

-        In the introduction the SNP allele variants and their functional roles should be introduced.

Methods and Results

-        It is one of the strong parts of the manuscript, these excellently described

Discussion

·       Include a section on strengths / limitations.

·       What mechanisms of action support these findings?

·       What does this article contribute to, the authors should make their own assessment and include their own discussion of the results shown in the manuscript?

Conclusion

In the Conclusion section, state the most important outcome of your work. Do not simply summarize the points already made in the body — instead, interpret your findings at a higher level of abstraction. Show whether, or to what extent, you have succeeded in addressing the need stated in the Introduction (or objectives). 10.3390/ijms231911846

Author Response

We thank you for the valuable and constructive comments. We have carefully reviewed the comments, and the manuscript was revised according to the suggestions provided by you. Our detailed responses are in the attached file. We have marked the modifications we made in yellow. The revised version of the manuscript has been approved by all authors.

We hope that the revised version is suitable for publication. We look forward to hearing from you, and we would be happy to respond to any other question or comment that you may have.

Reviewer 2 Report

The manuscript "Polymorphism in drug transporter gene ABCB1 is associated with drug response in Saudi epileptic pediatric patients" by  et al. presents an investigation of the association between single nucleotide polymorphisms (SNPs) of the ATP-binding cassette subfamily B member 1 (ABCB1) gene in epileptic pediatric patients and their response to anti-seizure medications (ASMs). The study is interesting considering the pediatric patients and the results seem to be supported by the findings. Although the number of patients is rather low, there are a couple of clear correlations between good and bad responders with ABCB1 gene SNPs and haplotypes analyzed.

Overall the manuscript is well-written and the results are presented clearly. Considering that this study is the first to investigate this association, I reckon that it may be of interest to the readers of Biomedicines. Although I'm not an expert in the field, I find that this study could inspire further investigations in the field, underlying another pathological state where personalized treatment is required.

I have only a couple of minor suggestions, therefore, I can suggest its publication in Biomedicines.

1. The introduction is rather poor. The authors could provide more information about the choice and function of ABCB1, including more investigations related to epilepsy and SNPs.

2. The exact type of t-test employed should be given (e.g. unpaired, two-tailed t-test)

Language is fine

Author Response

(The authors gave the same response as above.)

Reviewer 3 Report

Magadmi et al. studied the associations between three single-nucleotide polymorphisms of the ABCB1 gene with drug responsiveness or drug resistance in a cohort of paediatric epileptic patients treated with antiseizure medications in Saudi Arabia. Unfortunately, as presented, the manuscript is not publishable. To improve the presentation of the manuscript/data, I have many suggestions as listed below (irrespective of the hierarchy of importance). The authors must refer to a professional editing service to help them revise the text of the manuscript, reedit the manuscript, reorganise the manuscript text for logical presentation and context, and for cohesion. The manuscript will benefit from a major revision and major rewriting. Please refer to the attached PDF for additional comments and tips.

1.      The authors mention that more than 20 antiseizure medications (ASMs) are available, and that a third of epilepsy patients are affected by resistance to ASMs. The ABCB1 product, P-glycoprotein is thought to generally mediate resistance to many types of drugs because it functions as a promiscuous drug-efflux pump. P-glycoprotein is also shown to efflux some ASMs (PMIDs 26421491, 34610620, 32890316, some already cited by the authors). Because the size of the study cohort is small, the authors can easily obtain and present information on the treatment regimens used for individuals enrolled in their study cohort. For example, the authors can group the patients according to the type of ASMs prescribed and explore the genetic and SNP associations with resistance to a certain type of prescribed ASM. Presenting such detailed analyses will strengthen the authors argument about individualized medicinal therapy because associating resistance to a certain ASM with a specific SNP will be more meaningful and clinically applicable than a general SNP study with no information on the ASM prescriptions/regimens. Additionally, some of the inconstancies observed for different SNPs may be explained by the type of medication prescribed or by the combinations of SNPs for some individuals. Please revise the study with such considerations and provide the additional treatment data to be analysed in the context of SNPs.

2.      Some study rationales are not clearly presented or are missing altogether. For example, nowhere in the introduction section, the authors mention the rationale for studying a sample of the Saudi paediatric patients (see lines 56–60).

3.      The rationale for studying the liver-function test is missing or clear link is missing between the study aims and liver-function tests. How does the liver-function test link with P-glycoprotein SNPs in the context of epilepsy? The rationale should be clarified in the introduction and a detailed discussion must be presented on the LFT results.

4.      In line 187, ‘consanguinity’ was mentioned but this concept or the rationale behind it in the context of epilepsy was not explained. How was consanguinity reported and recorded in the study? Does this mean that a third of the patients in the study shared common parents, i.e., were siblings? How was this possible if the study was a multi-centre study? Were all the siblings in the study referred to the same hospital? These questions need to be clarified in the text. The authors must also refer to those studies that have shown or proven the relationship between consanguinity and the risk of epilepsy.

5.      The authors should explain the models of inheritance used in their study. (See lines 250 and 253).

6.      The ‘Saudi’ attribution was mentioned in the title and abstract but not much everywhere else.

7.      Methods must be rewritten, summarised, and revised. For example, reporting ‘rpm’ for centrifugation is not informative in an academic manuscript; the authors must report the g-force so the study can be replicated if required. Some methods statements are too detailed or written using repetitive joining words, e.g., ‘then’. This makes reading the text tedious and tiring for the audience of the paper.

8.      Some methodological choices must also be justified. For example, was the fidelity of the polymerase enzyme considered? What if a PCR polymerase introduced mutations into the investigated sequences? What is the evidence that the enzyme used is the best polymerase and did not cause aberrations?

9.      In Table 1, in methods, the so-called "AA change location" or "protein change location" do not make sense and much of the information is unmatched. What is the standard nomenclature system describing such changes? See the attached PDF with comments.

10.  The discussion is not presented in a logical order but seems to follow how the results were presented by the authors. Also, much of the information that is more appropriate to be presented in the introduction section have been presented in the discussion. The authors need to rewrite the introduction and the discussion section and reorganise the logical flow of the text.

11.  The discussion suffers from inconsistent formatting (e.g., reintroduction of some abbreviations, or lack abbreviation definitions and inconsistent citation formatting). At the end of the discussion, the authors extrapolate on the expression of P-glycoprotein; however, the presented data do not contain any evidence on P-glycoprotein expression.

12.  The authors stated the limitations of their study in the conclusion. These are to be best placed in the discussion. The conclusion should highlight the most important findings of the research and the significance of the study.

13.  Some English problems include excess nominalisations, use of long strings of nouns as noun trains, repetitions, informal language (contractions of verbs), and vague or unclear expressions ("our population", "the SNP"). An appointed editor should rewrite the text considering these problems. We have provided some editorial tips in the attached PDF; however, the mistakes are too many to list here in this review and is beyond the scope of unremunerated work by a peer-reviewer.

14.  I dispute the evidence and the claim preceding reference 34 (line 439). The reference 34 may not be the correct one for this claim. The authors must revise this.

See the overall comments. Refer to the attached PDF for some exemplary comments.

Author Response

(The authors gave the same response as above.)

Reviewer 4 Report

This is a very interesting paper investigating the impact of pharmacogenetics of ABCB1 in drug response in Saudi epileptic pediatric patients. The paper is well written, and of interest for the journal. However, several minor changes are recommended. The authors have done a great effort in improving the paper. I only recommend minor changes.

INTRODUCTION

1- The introduction may include not only the explanation for the P-glycoprotein, but also for CYP enzymes, because antiepileptic drugs are also metabolized by CYP 450 enzymes. 

2- At the end of the introduction, the authors should justify why they have selected these polymorphisms to investigate associations with drug response.

MATERIAL AND METHODS

1- I recommend to join the subsection "study design" with "participant recruitment and classification". A potential name for the new section could be "Participants and study design".

DISCUSSION

I recommend to add a limitations and strentghs subsection at the end of the discussion (lines 456-463).

Author Response

Thank you very much for taking the time to review this manuscript. Please find the detailed responses in the attached file and the corresponding revisions/corrections highlighted/in track changes in the re-submitted files.

Reviewer 5 Report

In the manuscript by Rania Magadmi and colleagues titled "Polymorphism in drug transporter gene ABCB1 is associated with drug response in Saudi epileptic pediatric patients.". Epilepsy is one of the most prevalent chronic neurodisorders in children. There are more than 20 anti-seizure medications (ASMs) available on the market, yet one-third of epileptic patients are still affected by drug-resistant epilepsy. Therefore, the purpose of this study was to investigate the association between single nucleotide polymorphisms (SNPs) of the ABCB1 gene in epileptic pediatric patients and their response to ASMs. In Jeddah, Saudi Arabia, a multicentric, cross-sectional study was conducted among Saudi children with epilepsy. With the help of Sanger sequencing, we genotyped ABCB1 polymorphisms at exon 12 (rs1128503), exon 21 (rs2032582), and exon 26 (rs1045642). In this study, 85 children with epilepsy were enrolled: 43 showed a good response to ASMs, while 42 showed a poor response. In contrast to poor responders, good responders were significantly more likely to have TT genotypes at rs1045642 and rs2032582 SNPs. Additionally, haplotype analysis revealed that the T-G-C haplotype frequency at rs1128503, rs2032582, and rs1045642 was only present in poor responders. In conclusion, this study represents the first pharmacogenetic analysis of ABCB1 in Saudi epileptic children and demonstrates a significant association between rs1045642 and rs2032582 variants and patient response. Despite the small sample size, the results demonstrate the importance of individualized treatment for epileptic patients. Regarding the present manuscript, I would like to make a few comments.

-This is my first time making comments on a revised manuscript

-Similar studies have shown that the ABCB1 gene is associated with the effects of different drugs, maybe more information about that is required.

-Perhaps the importance of the treatment change should be highlighted in the introduction

-The study was only conducted for a few months, therefore the message of the study will be more powerful if there are more patients involved

-Perhaps the statistical method misses the two-tailed distribution, as shown in a further table.

-The results are well written, although Figure 1 and Table 7 do not appear to be of major importance to the manuscript.

-The authors did not discuss the key information they found in their study, the significance of two SNPs and the importance of the recessive effect.

-It is my first time reading this manuscript and the authors have made changes based on the opinions of other reviewers. The major limitation of the study is the number of subjects evaluated.

Author Response

(The authors gave the same response as above.)

Round 2

Reviewer 3 Report

Second round of review:

I thank the authors for considering the peer-review feedback that I provided during the initial round of review. Unfortunately, the authors seem to be utterly unwilling to spend sufficient time to implement the suggestions for the betterment of their own manuscript (this is surprising as they did not find any value in the positive criticism I provided). The authors have been dismissive in applying many appropriate suggestions despite the immense amount of time I spent providing feedback. Their claim that the other peer-reviewers have congratulated them for writing the paper does not mean that the other two peer-reviewers have provided the best peer-review they could. I provide some additional feedback to clarify and reinforce the first peer-review I provided, and afterwards, it is up to the authors to accept or implement these suggestions or be dismissive again. It will not affect the publisher as they would be happy to take the APC and publish the paper. The scientific community will not benefit.

1.      Please note that the conclusion has not been written in a logical way. Many statements in the conclusion must be placed in the introduction to the study to help clarify the strengthen the rationales of the study. For example, revise the statements in lines (305–313, 318–325) and consider rewriting them in the introduction section.

2.      Editorial suggestions provided in the attached PDF during the first round of peer-review were not implemented. For example, see the usage of the words ’90-voltage’ (line 149) and ‘hypnosis’ instead of hypothesis (349). Refer to the annotated PDF file or present it to the editor who has or will edit this manuscript. English writing can be improved immensely, especially, in the methods sections where the word ‘then’ has been repeated. Writing a better text will please the readers of the text and benefit them as well as benefit the authors. At least that is how I believe, contrary to the other two congratulatory reviewers.

3.      In line 331, the authors mention that their data support the hypothesis that changes in the expression of Pgp at the blood–brain barrier can hinder ASMs from reaching a high concentration. However, this statement is factually wrong because the authors have not studied the expression of this protein at the barrier and have provided no protein evidence. The authors must revise their study for factual accuracy and do not inflate their conclusions.

4.      SNPs do not need to be written in parentheses. Revise the text and write all the SNPs without parentheses like those written in lines 28, 29, and 32. Consistency is appreciated; inappropriate punctuation is unnecessary. Also note that use of colon in line 90 is unnecessary. Delete.  

5.      Reference 35 is not correct for the claim that Pgp is expressed at higher levels in the blood–brain barrier than in the liver. I pointed out this lack of fact-checking in the initial review; however, the authors have not corrected this mistake. See here: https://www.proteinatlas.org/ENSG00000085563-ABCB1/tissue#rna_expression as was included in the annotated PDF file during the initial review.

6.      In Figure 1, change the word ‘Project’ to ‘Study Cohort’ in all the figure panels.

Thank you and good luck. 

I thank the authors for considering the peer-review feedback that I provided during the initial round of review. Unfortunately, the authors seem to be utterly unwilling to spend sufficient time to implement the suggestions for the betterment of their own manuscript (this is surprising as they did not find any value in the positive criticism I provided). The authors have been dismissive in applying many appropriate suggestions despite the immense amount of time I spent providing feedback. Their claim that the other peer-reviewers have congratulated them for writing the paper does not mean that the other two peer-reviewers have provided the best peer-review they could. I provide some additional feedback to clarify and reinforce the first peer-review I provided, and afterwards, it is up to the authors to accept or implement these suggestions or be dismissive again. It will not affect the publisher as they would be happy to take the APC and publish the paper. The scientific community will not benefit.

1.      Please note that the conclusion has not been written in a logical way. Many statements in the conclusion must be placed in the introduction to the study to help clarify the strengthen the rationales of the study. For example, revise the statements in lines (305–313, 318–325) and consider rewriting them in the introduction section.

2.      Editorial suggestions provided in the attached PDF during the first round of peer-review were not implemented. For example, see the usage of the words ’90-voltage’ (line 149) and ‘hypnosis’ instead of hypothesis (349). Refer to the annotated PDF file or present it to the editor who has or will edit this manuscript. English writing can be improved immensely, especially, in the methods sections where the word ‘then’ has been repeated. Writing a better text will please the readers of the text and benefit them as well as benefit the authors. At least that is how I believe, contrary to the other two congratulatory reviewers.

3.      In line 331, the authors mention that their data support the hypothesis that changes in the expression of Pgp at the blood–brain barrier can hinder ASMs from reaching a high concentration. However, this statement is factually wrong because the authors have not studied the expression of this protein at the barrier and have provided no protein evidence. The authors must revise their study for factual accuracy and do not inflate their conclusions.

4.      SNPs do not need to be written in parentheses. Revise the text and write all the SNPs without parentheses like those written in lines 28, 29, and 32. Consistency is appreciated; inappropriate punctuation is unnecessary. Also note that use of colon in line 90 is unnecessary. Delete.  

5.      Reference 35 is not correct for the claim that Pgp is expressed at higher levels in the blood–brain barrier than in the liver. I pointed out this lack of fact-checking in the initial review; however, the authors have not corrected this mistake. See here: https://www.proteinatlas.org/ENSG00000085563-ABCB1/tissue#rna_expression as was included in the annotated PDF file during the initial review. Reference 35 does not provide any evidence to support that Pgp is expressed more in the BBB than in the liver.

6.      In Figure 1, change the word ‘Project’ to ‘Study Cohort’ in all the figure panels.

7.      The authors must revise the use of the abbreviations. LD for example, comes out of the blue and has not been defined (lines 424 and 428).

Thank you and good luck. 

Author Response

(The authors gave the same response as above.)

Reviewer 5 Report

I would like to thank the reviewers for taking the time to respond to my previous comments. There are no further comments needed from my side as the manuscript now reads well.

Round 3

Reviewer 3 Report

I thank the authors for revising the text. The manuscript is now publishable though many sentences can be rephrased or rewritten to improve the language expression or achieve brevity. I suggest leaving these to the editorial, typesetting, and proofreading stages. 

I thank the authors for revising the text. The manuscript is now publishable but still not in its presented form. Many sentences can be rephrased or rewritten to improve the language expression and achieve brevity. Word economy is important for academic expression and publications. I suggest leaving these to the editorial, typesetting, and proofreading stages because I do not have sufficient time to provide another round of feedback.